# CPSF3 Promotes Pre-mRNA Splicing and Prevents CircRNA Cyclization in Hepatocellular Carcinoma

**DOI:** 10.3390/cancers15164057

**Published:** 2023-08-11

**Authors:** Ying Huang, Haofei Ji, Jiani Dong, Xueying Wang, Zhilin He, Zeneng Cheng, Qubo Zhu

**Affiliations:** 1Xiangya School of Pharmaceutical Sciences, Central South University, Changsha 410013, China; huanghy@csu.edu.cn (Y.H.); 217211052@csu.edu.cn (H.J.); dongjn0630@163.com (J.D.); 217211047@csu.edu.cn (Z.H.); chengzn@csu.edu.cn (Z.C.); 2China National Intellectual Property Administration, Beijing 100088, China; wangxueying@cnipa.gov.cn

**Keywords:** competing endogenous RNA (ceRNA), circular RNAs (circRNAs), cleavage and polyadenylation specificity factor (CPSF), hepatocellular carcinoma (HCC), JTE-607

## Abstract

**Simple Summary:**

CircRNAs are involved in the tumorigenesis and metastasis of hepatocellular carcinoma (HCC). Previous studies revealed that circRNA expression was associated with 3′-end splicing. As the core executor of 3′-end cleavage, CPSF3 should modulate the circularization of circRNAs. From our research, we found that CPSF3 was highly expressed in HCC cells, and a high CPSF3 level was predictive of a poor prognosis. The increase of CPSF3 promoted the conversion of pre-mRNA processing products from circRNA to linear mRNA, thus inducing the tumorigenesis and metastasis of HCC. JTE-607, which is a chemical inhibitor of CPSF3, exerted a therapeutic effect on HCC. This elaborate work not only sheds a light on the research about the initiation and progression of HCC, but also contributes to the diagnosis and treatment of HCC.

**Abstract:**

CircRNAs are crucial in tumorigenesis and metastasis, and are comprehensively downregulated in hepatocellular carcinoma (HCC). Previous studies demonstrated that the back-splicing of circRNAs was closely related to 3′-end splicing. As a core executor of 3′-end cleavage, we hypothesized that CPSF3 modulated circRNA circularization. Clinical data were analyzed to establish the prognostic correlations. Cytological experiments were performed to determine the role of CPSF3 in HCC. A fluorescent reporter was employed to explore the back-splicing mechanism. The circRNAs regulated by CPSF3 were screened by RNA-seq and validated by PCR, and changes in downstream pathways were explored by molecular experiments. Finally, the safety and efficacy of the CPSF3 inhibitor JTE-607 were verified both in vitro and in vivo. The results showed that CPSF3 was highly expressed in HCC cells, promoting their proliferation and migration, and that a high CPSF3 level was predictive of a poor prognosis. A mechanistic study revealed that CPSF3 enhanced RNA cleavage, thereby reducing circRNAs, and increasing linear mRNAs. Furthermore, inhibition of CPSF3 by JET-607 suppressed the proliferation of HCC cells. Our findings indicate that the increase of CPSF3 in HCC promotes the shift of pre-mRNA from circRNA to linear mRNA, leading to uncontrolled cell proliferation. JTE-607 exerted a therapeutic effect on HCC by blocking CPSF3.

## 1. Introduction

Hepatocellular carcinoma (HCC) is the fourth most common cause of cancer-related death worldwide and is the most prevalent primary liver malignancy. It is mainly caused by the deregulation of gene networks that maintain cellular homeostasis in hepatocytes [1,2]. As non-coding RNAs (ncRNAs) affect gene networks at both the transcriptional and post-transcriptional levels [3,4], several ncRNAs have been found to play a crucial role in the carcinogenesis and metastasis of HCC [5,6,7]. Circular RNAs (circRNAs), a type of covalently closed ncRNA, have recently emerged as potential molecular markers and therapeutic targets for HCC. They can influence cellular physiology through various molecular mechanisms [8,9,10], including acting as competitive endogenous RNAs (ceRNA) that sponge miRNAs [11], and protein-binding RNAs that modulate gene expression [12]. Additionally, circRNAs are considered more stable biomarkers and therapeutic targets compared to other RNAs because they lack a 5′-cap and 3′-poly(A) tail, which results in them being resistant to degradation by the exonuclease RNase [13].

Previous studies have shown that circRNAs are universally reduced in HCC tissues [14], but the exact molecular mechanism underlying the inhibition of circRNA formation in tumors remains unknown. CircRNAs differ from linear RNAs in that they generate covalently connected 5′ splice acceptor and 3′ splice sites in reverse without polyadenylation [15]. Although circRNAs originate from internal exons of protein-coding genes, they generally have short exon spans and often contain early transcriptional exons [16]. Bioinformatic analysis has revealed that circRNAs are comprised of few 3′-untranslated region (3′-UTR) sequences and a significantly lower density of PAS sequences compared to their corresponding mRNAs (Appendix B Figure A1, and Appendix A). This suggests that the formation of circRNA is affected by the RNA 3′-end cleavage and polyadenylation process, and downregulation of circRNAs in HCC may be caused by abnormal RNA splicing and maturation.

The two processes of 3′-end cleavage and polyadenylation play a pivotal role in the maturation of precursor RNA and are closely linked to two subsequent processes of alternative splicing (AS) and alternative polyadenylation (APA) of RNA [17,18]. These processes are regulated by the interaction of specific sequences within precursor RNA with the 3′-end formation complex, which is composed of 4 multi-subunit complexes and a total of 15 proteins [17]. CPSF is the core component for RNA cleavage and recognizes the PAS specifically [19]; CFIm forms a dimer to bind with two UGUA sequences upstream of the PAS, resulting in enhanced polyadenylation site cleavage [5]; CFIIm does not directly interact with RNA, but is involved in the polyadenylation process [20]; additionally, CSTF is necessary for RNA cleavage and plays a crucial role in interacting with the U/GU-rich regions downstream of the PAS [21].

Recent research has indicated that the protein complexes and snRNAs that play a role in the back-splicing of circRNA are similar to those involved in the processing and maturation of linear RNA via canonical splicing. These similarities lead to a competitive relationship between the biogenesis of circRNAs and mRNAs. Protein-coding genes switch from mRNA to circRNA when pre-mRNA processing is blocked [22,23]. In previous studies, we have found that pre-RNA processing factors, such as NUDT21, which is involved in 3′-end formation, are expressed at abnormal levels in HCC, leading to uncontrolled tumor proliferation [5,6,14]. Therefore, we sought to investigate whether the core executor for PAS site cleavage, CPSF, plays a significant role in circRNA biogenesis and HCC tumorigenesis. The CPSF complex is composed of six proteins (CPSF1, CPSF2, CPSF3, CPSF4, FIP1L1, and WDR33); only CPSF2 and CPSF3 can catalytically cleave RNA with endonuclease activity [24], while the other four proteins form a stable tetramer that specifically binds to the PAS [25]. CPSF3 has been implicated in various cancers [26], including lung cancer [27], colorectal [28], prostate [29], and triple-negative breast cancers [30]. It has even been implicated as a potential therapeutic intervention target for acute myeloid leukemia (AML) and Ewing’s sarcoma (ES) using the inhibitor JTE-607 [31]. However, the mechanistic study of CPSF3 in HCC has not yet been conducted. Therefore, this research aims to analyze the correlation between CPSF3 and HCC, and to reveal the molecular mechanism of CPSF3 in tumorigenesis and metastasis by modulating the balance of circular and linear transcripts in HCC.

## 2. Materials and Methods

### 2.1. Bioinformatic Analysis

Sequence data of the 3′-UTR and PAS in circRNAs and mRNAs were downloaded from Arraystar Human CircRNA microarray. Information about circRNA expression was downloaded from the Gene Expression Omnibus (GEO) database (https://www.ncbi.nlm.nih.gov/geo/, accessed on 1 November 2021). Gene and miRNA expression data were obtained from either The Cancer Genome Atlas (TCGA) database (https://portal.gdc.cancer.gov/, accessed on 1 January 2022) or the GEO database. To predict the circRNA/miRNA interaction, CircInteractome (http://circinteractome.nia.nih.gov/, accessed on 1 May 2022) was utilized, and for predicting the miRNA/mRNA interaction, TargetScanHuman 8.0 (https://www.targetscan.org/vert_71/, accessed on 1 May 2022) was used.

### 2.2. Clinical Cancer Samples and Tissue Microarrays

The cancer samples used in this study were provided by the Third Xiangya Hospital in Central South University. These samples were collected and stored in liquid nitrogen until analysis. The study received ethical approval from the Ethics Committee of the Xiangya College of Pharmacy in Central South University (Ethical code: No. 202251, Appendix A).

Outdo Biotech Co., Ltd. (Shanghai, China) provided the tissue microarrays of HCC and the corresponding para-tumorous (PT) tissues. CPSF3 protein levels were determined by staining tissue microarray slides, which were then rated for staining color depth by three independent individuals. Only tissue samples were utilized from individuals who had not received anti-cancer medication prior to tumor removal. The TNM stage and overall survival data from all patients were included in Appendix A for analysis.

### 2.3. Cells Culture and Plasmid Transfection

The following cells were acquired from the Xiangya Experiment Center (Changsha, China): HepG2, Bel7404, Hep3B, Huh-7, Bel7402, and L02. These cells were cultured in either RPMI-1640 (VivaCell, Shanghai, China) or high-glucose DMEM (VivaCell) medium, which were supplemented with 10% fetal bovine serum (Cell-box, Changsha, China) and 1% penicillin–streptomycin (VivaCell). All cells were maintained at 37 °C in a cell culture incubator with 5% CO_2_, and the medium was changed every two to three days.

The CPSF3 sgRNA (Appendix A) was inserted into a lentiCRISPR v2 vector to create the CPSF3-knockout (KO) plasmids and the CPSF3 CDS sequence (NM 001321833) was inserted into a pcDNA3 vector to create the CPSF3-overexpressing (OE) plasmid. Plasmids were transfected into cells with the ViaFectTM transfection reagent (Promega, Madison, WI, USA). Stable transfected cells were selectively cultured in a medium supplemented with 2 µg/mL G418 (Macklin Biochemical, Shanghai, China) or 2 µg/mL puromycin (Bioss, Beijing, China). To generate CPSF3-KO monoclonal cells, stably transfected cells were seeded at a density of 1 cell per well in 96-well plates, and the monoclonal cell lines were then confirmed by genome sequencing and western blot.

### 2.4. RNA Sequencing

The RNA samples from the CPSF3-KO, CPSF3-OE, and control cells were extracted using TRIzol reagent (Invitrogen, Carlsbad, CA, USA), and then sequenced by Biomarker Technologies (Beijing, China) following the experiment protocol.

### 2.5. qRT-PCR

Total RNA was extracted from cultivated cells. The cDNA of circRNA and mRNA were synthesized using the RevertAid First Strand cDNA Synthesis Kit (Thermo Fisher Scientific, Waltham, MA, USA), while the cDNA of miRNA was synthesized using the miRNA First Strand cDNA Synthesis (Tailing Reaction) Kit (Sangon Biotech, Shanghai, China). For quantification, real-time PCRs with the specified primers (Appendix A) were performed using Hieff qPCR SYBR Green Master Mix (Yeasen Biotech, Shanghai, China). The levels of mRNA and circRNA were normalized to β-actin, while the levels of miRNA were normalized to U6. Each experiment was repeated 3 times and the expression levels were assessed via the 2^−ΔΔCt^ method.

### 2.6. Dot Blot

Total RNA was extracted from cell cultures, and 5 µg of total RNA from each sample was added to the well of the dot blotting spotter. The RNA was adsorbed on a Hybond-N+ nylon membrane (Amersham, Little Chalfont, UK) under a vacuum, and then cross-linked to the membrane using UV. Before overnight hybridization at 35 °C with the biotin-labeled probes (Appendix A), the membranes were prehybridized for 2 h at 35 °C in the PerfectHybTM Plus Hybridized buffer (Sigma, St. Louis, MO, USA). The results displayed by the Detection Kit (Beyotime, Shanghai, China) were visualized by the chemiluminescence imager (Bio-Rad, Hercules, CA, USA).

### 2.7. Western Blot

The RIPA lysis buffer (Beyotime, Shanghai, China) containing protease inhibitor (Sigma, St. Louis, MO, USA) was employed for protein extraction, and the protein concentration was determined using the BCA kit (Bioss, Beijing, China). The extracted proteins were separated through electrophoresis on an 8% SDS-polyacrylamide gel, and subsequently transferred onto PVDF membranes (Immobilon^®^-P, Millipore, Billerica, MA, USA). The membranes were then incubated overnight at 4 °C with primary antibodies, including anti-CPSF3 (1:1000, 11609-1-AP, Proteintech, Wuhan, China), anti-E2F1 (1:1000, YT5811, Immunoway, Plano, TX, USA), anti-NCAPG2 (1:1000, YT2105, Immunoway), anti-SLC22A3 (1:1000, YT3232, Immunoway, Plano, TX, USA), anti-UBXN7 (1:1000, HPA049442, Altas Antibooies, Zurich, Switzerland), and anti-β-actin (1:5000, ab8227, Abcam, Cambridge, UK). Finally, the membranes were coated with an ECL reagent (Beyotime) and analyzed using a chemiluminescence system (Bio-Rad).

### 2.8. Cell Proliferation, Clone Formation, and Wound Healing Assays

To evaluate cell proliferation, a CCK-8 assay kit (biosharp, Beijing, China) was employed. Cells were seeded at a density of 1 × 10^3^ cells per well in a 96-well plate, with three wells per condition, and incubated for up to 5 days. Each well was treated with 10 µL of the CCK-8 reagent and incubated for 2 h at 37 °C. The absorbance at a wavelength of 450 nm was measured using a microplate reader to determine cell viability.

To assess long-term cell viability and proliferation, colony formation assays were conducted by growing cells in medium containing 0.35% soft agar (1000 cells/well) for 1–2 weeks until colonies formed. These colonies were fixed with 4% paraformaldehyde and stained with 0.1% crystal violet.

The cell migration abilities were investigated with wound healing assays. Cells were seeded in 6-well plates and scratched with 10 μL pipette tips. The scratches at the same location were photographed with a microscope at different time intervals to observe cell migration.

### 2.9. JTE-607 Studies

To measure dose response, 1000 cells were seeded per well on a 384-well plate. On the next day, JTE-607 was added to the wells at different concentrations, while 10% DMSO was used as the control. After 72 h of incubation, cell viability was measured using a CCK8 kit. Similarly, for the clone formation assay, 1000 cells were seeded per well in medium containing 0.35% soft agar and treated with different doses of JTE-607.

### 2.10. Xenograft Tumorigenic Assays

HepG2 cells (2–4 × 10^6^) were subcutaneously injected into the lateral backside of 16 male BALB/c nude mice (5 weeks old). The tumor size was measured every two days with calipers, and the volume was calculated using the formula volume = (1/2) × length × width^2^. The tumor site of the experimental group mice was administrated with 30 μL of JTE-607 at a concentration of 0.2 mg every two days, whereas the control group was injected with 30 μL of normal saline. After 4 doses, the mice were sacrificed to harvest the xenografts, which were further evaluated by weighing.

The study was conducted in accordance with the guidelines of the Animal Ethics Committee of Central South University, and all necessary measures were implemented to ensure animal welfare (Ethical code: No. 202251, Appendix A).

### 2.11. Statistical Analysis

Statistical analysis was performed using the R software package (R Version 3.5.2), SPSS 16.0, and GraphPad Prism 8 software package. The determination of statistical significance was conducted through Student *t*-test (between two groups) or ANOVA (more than two groups) followed by the post hoc Bonferroni test. The assessment of correlations was carried out using Spearman correlation coefficients. Kaplan-Meier plot was used to determine the patient’s survival. Furthermore, the logistic regression model was applied to estimate the relationship between CPSF3 and HCC risk. The Cox analysis was applied to validate the associations among CPSF3 expression, survival, and other clinical features.

All *p* values were 2-sided, and differences with *p* < 0.05 were considered statistically significant (* *p* < 0.05, ** *p* < 0.01, *** *p* < 0.001). All data were presented as mean ± standard deviations (s.d.).

## 3. Results

### 3.1. The Downregulation of CircRNAs in HCC Was Dependent on the PAS Sequence

There is a strong preference for including 5′-end exons in the production of circRNA, whereas 3′-UTR sequences are typically excluded [16]. Through the analysis of 5393 circ-RNAs from the Agilent-069978 Arraystar Human CircRNA microarray (Appendix A), we observed a significantly lower proportion of 3′-UTR sequences within circRNAs compared to their corresponding mRNAs (Appendix B Figure A1a). This finding was further supported by the observation that circRNAs had a significantly lower density of PAS, which is the primary regulatory element responsible for determining the 3′-UTR length, when compared to mRNAs (Appendix B, Figure A1b).

To further explore the effect of PAS on circRNA expression in HCC, two publicly available datasets, GSE94508 [32] (five pairs of HCC and PT tissues, Appendix A) and GSE97332 [33] (seven pairs of HCC and PT tissues, Appendix A) were analyzed. The results revealed a significant decrease in circRNA expression in HCC tissues compared to PT tissues (Appendix B, Figure A1c). To investigate the sequence specificity of the global downregulation of circRNAs, the circRNAs were classified into two groups based on the presence or absence of PAS. The data showed that the expression of circRNAs with PAS sites was much lower than that of circRNAs without PAS sites. Interestingly, circRNAs with PAS sites exhibited a more pronounced decrease in HCC tissues (Appendix B, Figure A1d), indicating that circRNA biogenesis is dependent on the PAS sequence, and the downregulation of circRNAs in HCC could be attributed to aberrant 3′-end splicing and polyadenylation processes.

### 3.2. The 3′-End Formation Complex Was Associated with CircRNA Biogenesis and the Survival of HCC Patients

Because the 3′-end formation complex is essential for 3′-end splicing and polyadenylation, we wondered whether abnormal expression of the 3′-end formation complexes would influence circRNA biogenesis in HCC. The 3′-end formation complex consists of 15 proteins organized into 4 multi-subunit complexes, as shown in Figure 1a. We compared the mRNA expression levels of these proteins in 212 HCC and 50 PT tissues from the TCGA dataset and found that CPSF1, CPSF3, CPSF4, CPSF6, and CSTF2 were significantly upregulated in HCC (*p* < 0.001, fold change > 2, Figure 1b). Furthermore, patients with high levels of CPSF2, CPSF3, CPSF4, CPSF6, CSTF2, and SYMPK had shorter overall survival (OS) (*p* < 0.01, hazard ratio (HR) > 1.5, Figure 1c), according to survival analysis of 364 patients from TCGA dataset. We also analyzed the data from the GSE138734 database and found that CPSF3, CSTF2, CSTF3, and SYMPK were significantly inversely associated with the total expression of circRNA (*p* < 0.001, spearman correlation <−0.2, Figure 1d). In summary, many components of the 3′-end formation complexes were highly expressed in HCC and were associated with a decrease in total circRNA expression in tumors as well as a lower survival rate among patients, especially for CPSF3 and CSTF2. Considering that CPSF is the main executor for PAS cleavage, our research focused on the mechanism of CPSF3 in regulating the biogenesis of circRNA in HCC.

### 3.3. High Expression of CPSF3 Predicted Poor Prognosis in HCC Patients

To investigate the connection between CPSF3 and HCC, we first analyzed data obtained from TCGA and the GEO databases. By comparing the level of CPSF3 mRNA in 50 matched HCC tissue pairs from TCGA, we discovered that the expression in HCC tissues was significantly higher than that in PT tissues (*p* < 0.001). Likewise, the CPSF3 level in peripheral blood mononuclear cell (PBMC) samples from the GSE49515 microarray database was considerably higher in HCC patients compared to healthy individuals (Figure 2a). Additionally, Kaplan–Meier survival analysis was conducted, which revealed that patients with high CPSF3 expression had significantly shorter OS compared to those with low expression, and the hazard ratio between the two groups was 1.7 (*p* < 0.01, Figure 2b). Further analysis was performed by logistic regression to scrutinize the relationship between the CPSF3 level and the clinical pathological parameters of HCC, and the results demonstrated that CPSF3 was significantly related to tumor stage, tumor grade, and primary tumor size (Figure 2c). The univariate analysis showed that the OS of HCC patients was linked to tumor stage, primary tumor size, and high CPSF3 expression (*p* < 0.001). Furthermore, the multivariate Cox regression analysis indicated that the CPSF3 level was an independent prognostic factor for OS (*p* < 0.0001, Figure 2d). Due to the insufficient data, stage is not an independent risk factor for OS in multivariate analysis. A limitation with the GEO and TCGA in HCC is the oversampling of advanced stage disease. To corroborate our findings in early-stage disease, the expression of CPSF3 in early HCC was analyzed. Through logistic regression, we found T2 vs. T1 in primary tumor (total (N) = 280), the odds ratio in CPSF3 was 2.44 (95% CI: 1.46–4.14, *p*-Value = 0.0008); in tumor stage, II vs. I (total (N) = 262), the odds ratio in CPSF3 was 2.4 (95% CI: 1.40–4.15, *p*-value = 0.0015). According to these results, advance-stage tumors were more likely to have elevated CPSF3 expression than early-stage disease, which indicated that CPSF3 might be involved in the early development of HCC.

Since the correlation between transcriptome expression and protein expression was unclear, our bioinformatic results were validated through the analysis of clinical specimens. The western blot analysis showed that CPSF3 was expressed more highly in HCC tissues than in PT tissues (Figure 2e). Additionally, CPSF3 exhibited higher expression levels in several HCC cell lines (Huh7, Bel7402, Bel7404, HepG2, and Hep3B) compared to the normal liver cell line L02 (Figure 2f). Among these five HCC cell lines, HepG2 had the highest expression of CPSF3, while Bel7404 had the lowest. Tissue microarrays composed of samples obtained from 75 HCC patients were immune stained with the CPSF3 antibody (Appendix A), and the results confirmed the upregulation of CPSF3 in HCC. The results revealed that CPSF3 was highly expressed in 50 out of 75 tumor tissues, moderately expressed in 25 tumor tissues, and minimally expressed in most PT tissues (Figure 2g). The clinical significance of CPSF3 expression was evaluated in this cohort of 75 patients, and correlation analysis verified that CPSF3 level was significantly correlated with primary tumor factors and clinical stage (Figure 2h). Variable expression of CPSF3 in PT tissues might be associated with specific HCC etiologies. However, due to the limited sample size of tissue microarray data, we were unable to determine whether hepatic background CPSF3 expression was associated with a specific etiology such as cirrhosis.

All these data indicate that high expression of CPSF3 predicts poor prognosis in HCC patients, which suggests that CPSF3 may be involved in the tumorigenesis and metastasis of HCC.

### 3.4. CPSF3 Promoted the Proliferation and Migration of Hepatocellular Carcinoma Cells

To confirm the direct effect of CPSF3 on the proliferation and migration of HCC cells, CPSF3 knockout (KO) and overexpression (OE) cell lines were established in HepG2 and Bel7404 cells (Figure 3a). The growth rate, clone formation ability, and migration rate were examined in HCC cells with varying levels of CPSF3. Results from the CCK8 assay in both Bel7404 and HepG2 cells demonstrated that CPSF3-KO cells grew at a slower rate than the control cells. Conversely, CPSF3-OE cells exhibited an accelerated growth rate (Figure 3b). Results from the soft agar assay for anchorage-independent cell growth also showed that fewer and smaller colonies were generated in CPSF3-KO cells than in the control cells. In contrast, a greater number of larger colonies were generated in CPSF3-OE cells (Figure 3c). The wound healing assay was used to test the cell migration ability, and the data indicated that the healed area was reduced in CPSF3-KO cells but increased in CPSF3-OE cells (Figure 3d). All these results unequivocally demonstrate the crucial role of CPSF3 in cell growth and migration rate of HCC cells.

### 3.5. The CPSF3 Protein and PAS Element Were Critical for CircRNA Cyclization

The data presented above suggest that CPSF3 promotes the carcinogenesis and metastasis of HCC, and its oncogenic role might be associated with the modulation of circRNAs. To comprehend the mechanism underlying the effects of CPSF3 protein and PAS sequences on circRNA formation, a dual fluorescent reporter system incorporating Red/GFP was employed to study back-splicing. The Red-cGFP plasmid transcribed mRNA for the Ds-red protein followed by an IRES-driven circGFP circRNA. The Red-PAS-cGFP plasmid was generated by inserting an AAUAAA element in front of the circGFP in the Red-cGFP plasmid (Figure 4a). Confocal microscopy depicted a significant decrease in the green fluorescence intensity of Red-PAS-cGFP-transfected cells compared to Red-cGFP-transfected cells, while the red fluorescence intensity showed a remarkable increase (Figure 4b left), which implies that pre-mRNA comprising the PAS element preferentially matured into linear RNA instead of circRNA. This outcome was confirmed by real-time PCR, as the level of circGFP was substantially reduced while the level of Ds-red mRNA was significantly increased in Red-PAS-cGFP-transfected cells compared to Red-cGFP-transfected cells (Figure 4b right). To investigate the role of CPSF3 in circRNA biogenesis, the Red-PAS-cGFP plasmid was then transfected into Bel7404 cells with different levels of CPSF3 expression. Confocal microscopy revealed that the green fluorescence intensity of CPSF3-KO cells was higher than that of control cells, while the green fluorescence intensity of CPSF3-OE cells was lower than that of control cells. Nonetheless, the red fluorescence remained unchanged (Figure 4c left). Real-time PCR also showed that the ratio of the circGFP level to Ds-red mRNA level was higher in CPSF3-KO cells and lower in CPSF3-OE cells (Figure 4c right). These results indicate that CPSF3 enhances the maturation of pre-mRNA into linear mRNA and suppresses circRNA biogenesis by interacting with the PAS element, which is required for circRNA cyclization.

### 3.6. CPSF3 Promoted the Pre-mRNA Shift from CircRNA to Linear mRNA and Disrupted miRNA-Mediated Gene Silencing

The real effect of CPSF3 on circRNA expression in HCC cells was tested by analyzing the total RNA fractions obtained from CPSF3-KO HepG2 cells, CPSF3-OE Bel7404 cells, and negative control cells using high-throughput sequencing (Appendix A). The data obtained showed that CPSF3-KO cells had more types and a greater number of circRNAs compared to control cells, while CPSF3-OE cells had fewer types and a smaller number of circRNAs. The circRNAs affected by CPSF3 were mainly exonic, while intergenic and intronic circRNAs remained unchanged in all four cell lines (Figure 5a). The average level of each circRNA (unit: log2 SRPBM) increased from 5.8 ± 1.5 in control cells to 6.0 ± 0.19 in CPSF3-KO cells (*p* = 0.015), and conversely, decreased from 5.8 ± 1.3 in control cells to 5.4 ± 1.4 in CPSF3-OE cells (*p* = 3.1 × 10^−7^) (Figure 5b). Based on the obtained sequence data, it can be concluded that CPSF3 globally reduced the expression of circRNAs, which was consistent with the fluorescent reporter results.

To verify the sequence data, a number of circRNAs were chosen that fit the model. Firstly, 257 circRNAs that were either downregulated in CPSF3-OE cells or upregulated in CPSF3-KO cells (change in SRPBM > 1000) were selected from the RNA-seq data. Corresponding to these circRNAs, 14,077 mRNAs were also picked, which were either upregulated in CPSF3-OE cells or downregulated in CPSF3-KO cells (change in fpkm > 0.1). To supplement this data, the GEO database was trawled for 2048 downregulated circRNAs (*p* < 0.01) in HCC tissues. A total of 27 candidate circRNAs were identified as fulfilling these three requirements and were considered for validation. From these candidates, 15 circRNAs that contained the PAS elements in their pre-mRNAs were finally chosen (Figure 5c). Results from real-time PCR indicated that 13 of the 15 circRNAs were repressed in CPSF3-OE cells but overexpressed in CPSF3-KO cells (Figure 6a). The dot blot also confirmed that the circRNAs containing the PAS element, including hsa_circ_0001380 and hsa_circ_0078607, were lowly expressed in CPSF3-OE cells, yet were highly expressed in CPSF3-KO cells. However, hsa_circ_0008305, which did not contain the PAS element, was not affected by CPSF3 expression (Figure 6b). The corresponding mRNAs of hsa_circ_0001380 and hsa_circ_0078607 were UBXN7 and SLC22A3, respectively. Both the real-time PCR and western blot data showed that the corresponding protein-encoding genes were reduced in CPSF3-KO cells but elevated in CPSF3-OE cells, both at the RNA and protein levels (Figure 6c,d). All these data indicate that CPSF3 promotes the shift of pre-mRNA containing the PAS element from circRNA to linear mRNA in HCC cells.

We also tested for changes in the downstream pathways of circRNAs. Prior research has suggested that reducing circRNAs chronically results in the destabilization of the miRNAs bound to them [34]. CircRNAs act as miRNA sponges and protect them from exonuclease cleavage [35]. Universal downregulation of miRNAs has been observed in HCC [36]. As such, we selected four target miRNAs that were significantly downregulated in HCC for further study. Of these, miR-622 and miR-1247 are the targets of hsa_circ_0001380, while miR-188 and miR-1294 are the targets of hsa_circ_0078607 (Appendix B, Figure A2a,b). To confirm the binding between circRNAs and their target miRNAs, biotinylated circRNA probes were used to precipitate circRNAs and their target miRNAs. The results indicated that miR-622 and miR-1247 co-precipitated with hsa_circ_0001380, while miR-188 and miR-1294 co-precipitated with hsa_circ_0078607 in HepG2 cells (Figure 6e). The expression levels of these four miRNAs in cells with different CPSF3 levels were then detected by real-time PCR to check the influence of CPSF3 on downstream miRNAs. The data revealed that all four miRNAs were upregulated in CPSF3 KO cells, whereas they were downregulated in CPSF3-OE cells (Figure 6f), confirming that CPSF3-mediated circRNA reduction decreased their miRNA levels. As a tumor suppressor, miR-622 directly targets E2F1 [37]. A high E2F1 level is a risk factor for HCC [38]. MiR-188 has been found to target NCAPG2 directly, while increased NCAPG2 and decreased miR-188 levels have been associated with HCC progression and poor prognosis [39]. Bioinformatic analysis from the online tool TargetScan showed that the seed sequence of miR-622 bound with the 3′-UTR of E2F1, and the seed sequence of miR-188 bound with the 3′-UTR of NCAPG2 (Appendix B, Figure A2c). Data obtained from Gene Expression Profiling Interactive Analysis (GEPIA) demonstrated that these two target genes were upregulated in HCC, and their high levels indicated a shorter OS of HCC patients (Appendix B, Figure A2d,e). The silencing effect of miRNAs on their target genes was confirmed by the luciferase assay, as the hFluc/hRluc ratio decreased with the increase of miRNA-mimics in the luciferase reporter with the 3′-UTR of target genes, but no changes were detected in the luciferase reporter containing a mutant 3′-UTR (Figure 6g). To explore the effect of CPSF3 on these miRNA-targeted genes, the protein levels of E2F1 and NCAPG2 were detected by western blot. The results showed that both E2F1 and NCAPG2 were decreased in CPSF3-KO cells and increased in CPSF3-OE cells (Figure 6h). All of these findings demonstrated that CPSF3 reduced circRNA levels and disrupted miRNA mediated gene silencing in HCC cells.

### 3.7. Chemical Inhibition of CPSF3 by JTE-607 Inhibited the Proliferation of Hepatocellular Carcinoma Cells and Suppressed the Tumorigenicity in a Xenograft Mouse Model

JTE-607 has been recently identified as a small molecule that specifically inhibits CPSF3 and has been used as a new treatment for AML and ES. However, its impact on HCC remains unclear. Because CPSF3 suppression reduces HCC cell growth, we anticipate that the pharmacological suppression of CPSF3 by JTE-607 may constitute a unique treatment strategy for HCC. To test its sensitivity, multiple HCC cell lines and normal liver cells were subjected to a 72-h dose–response cell viability assay. The obtained CCK8 data showed that normal liver cells (L02, IC50 = 120.7 μM) and low CPSF3-expressing HCC cells (Bel7402, IC50 = 26.65 μM, Bel7404, IC50 = 50.97 μM) were not sensitive to JTE-607. In contrast, HCC cells with high CPSF3 expression displayed high sensitivity to JTE-607 (Huh7, IC50 = 0.53 μM, HepG2, IC50 = 0.14 μM) (Figure 7a). Therefore, all the drug efficacy tests were performed in high CPSF3-expressing HCC cells, such as Huh7 and HepG2. The colony formation assay results revealed that JTE-607 significantly reduced the colony formation ability of both Huh7 and HepG2 cells in a dose-dependent manner (Figure 7b). The RNA levels of CPSF3-regulated circRNAs (including hsa_circ_0001380 and hsa_circ_0078607), their targeting miRNAs (miR-622 and miR-188, respectively), and their corresponding mRNAs (UBXN7 and SLC22A3, respectively), were measured by real-time PCR. The data showed that JTE-607 treatment increased the levels of circRNAs and their target miRNAs, but decreased their corresponding mRNA levels in a dose-dependent manner (Figure 7c,d). Western blot analysis also showed that increasing the concentration of JTE-607 reduced the protein levels of miRNA-targeted genes and circRNA-corresponding genes (Figure 7e,f).

The efficacy of JTE-607 in inhibiting tumor growth was evaluated using a BALB/c xenograft mice model. The tumor volume curves (Figure 8a,b) indicate that JTE-607 considerably reduced the pace of tumor growth compared to the control group. After 20 days of growth, the weights of JTE-607 treated tumors were significantly lower as compared to the control tumors. Although hematoxylin and eosin staining showed no major tissue damage, immunohistochemical staining for Ki-67 revealed significant downregulation of its expression, thus suggesting that JTE-607 had an inhibitory effect on tumor proliferation (Figure 8c). We also monitored the body weights of the mice during the treatment and found that both groups showed stable growth trends without any significant differences (Figure 8d). Importantly, treatment with JTE-607 did not result in any increase in the serum levels of ALT, AST, BUN, or CREA, which are indicators of liver and kidney function; nor did it induce any changes in WBC, lymph, HGB, or PLT in the peripheral blood (Figure 8e,f). In conclusion, the results suggest that JTE-607 demonstrates excellent efficacy and safety when used as an anti-tumor agent for HCC in vivo.

## 4. Discussion

CircRNAs are a class of non-coding RNAs that are abundantly and stably expressed in various life forms [12], contributing to numerous cellular functions. These RNA molecules have been identified as significant players in various diseases, including cancer, and are being investigated as potential new biomarkers and therapeutic targets [40]. Prior research has shown that circRNAs are globally downregulated in HCC [14], although the mechanisms behind circRNA biogenesis in HCC are not yet fully understood. Analysis of their sequence revealed that circRNAs contain fewer 3′-UTRs and PASs compared to their corresponding mRNAs. In addition, circRNAs with PAS were found to be significantly reduced in HCC compared to those without PAS, indicating the involvement of 3′-end splicing and polyadenylation mechanisms in regulating circRNA biogenesis in HCC. Therefore, we conducted a comprehensive investigation to identify the proteins within the 3′-end complex, using bioinformatic analysis to determine their association with HCC. Our findings revealed that CPSF3, known for its critical role in PAS cleavage [24], was upregulated in HCC and strongly correlated with both patient survival and circRNA reduction. This observation was consistently supported by clinical specimens and HCC cell lines, whereby CPSF3 was found to be highly expressed in HCC cells, and its high expression indicated poor prognosis in HCC patients. Our cytological study further confirmed that CPSF3 facilitated the proliferation and migration of HCC cells. Through a mechanistic study, we discovered that CPSF3 promoted the maturation of pre-mRNA from circRNA to linear mRNA by interacting with the PAS element. High-throughput sequencing of CPSF3-OE and CPSF3-KO cells identified CPSF3-regulated circRNAs. The sequencing data were further confirmed by the molecular study, which showed that CPSF3 reduced the levels of hsa_circ_0001380 and hsa_circ_0078607, as well as disruption to their downstream miRNA-mediated gene silencing effect. In contrast, their corresponding linear mRNAs were increased by CPSF3, emphasizing the importance of CPSF3 in maintaining the balance between circular and linear transcripts. Because CPSF3 was found to promote the proliferation and migration of HCC cells by converting the output of circRNAs into protein-coding genes, we tested the antitumor effect of JTE-607, a CPSF3 inhibitor. Cytological and molecular studies demonstrated that HCC cells with high CPSF3 expression were sensitive to JTE-607, which suppressed cell growth by regulating CPSF3-regulated circRNAs, their target miRNAs, and their corresponding mRNAs. The xenograft mouse model demonstrated that JTE-607 significantly inhibited tumorigenicity in nude mice bearing HepG2 cells, indicating its potential as a therapeutic agent for HCC treatment. We have not yet developed a new dosage form for drug delivery. In the future, we plan to develop safer and more effective dosage forms and adopt safer delivery methods.

## 5. Conclusions

Our research proposes a novel mechanism of CPSF3 in HCC, whereby upregulation of CPSF3 in HCC cells enhances cleavage of pre-mRNA, leading to an increased abundance of linear transcripts and inhibition of circRNA formation. This ultimately reduces miRNA storage and results in the increased expression of target proteins, leading to uncontrolled proliferation of tumor cells. JTE-607 functions as an inhibitor of CPSF3 by blocking the cleavage of pre-mRNA and exerts its therapeutic effect by inhibiting the proliferation of HCC.

## Figures and Tables

**Figure 1 cancers-15-04057-f001:**
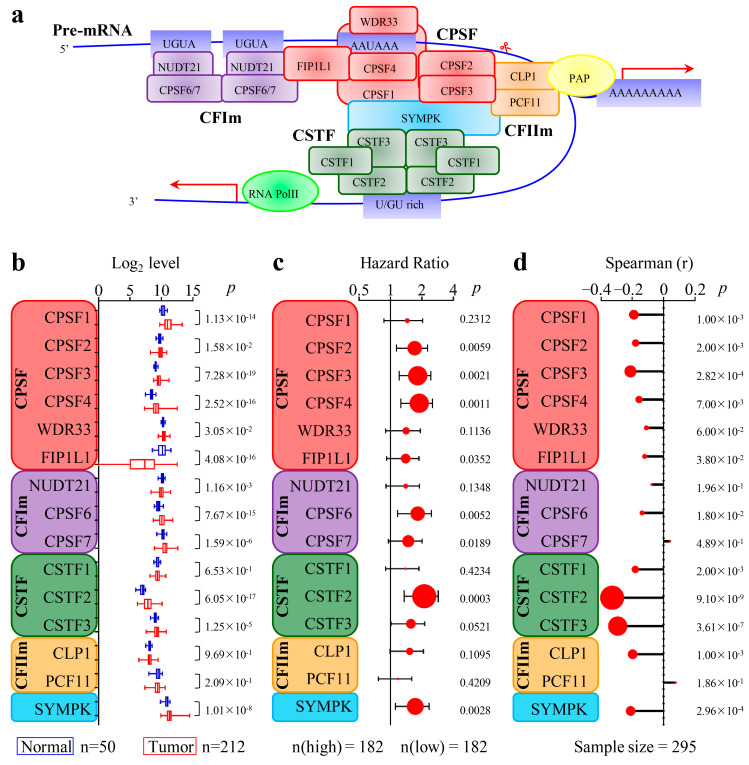
The effects of 3′-end formation complex components on HCC. (**a**) Illustration of the 3′-end formation complex. Scissor represents the cleavage site of CPSF. The arrow pointing right represents the direction of polyadenylation, while the arrow pointing left represents the direction of transcription. (**b**) The expression differences of each 3′-end formation complex component in HCC and PT tissues are compared using a box plot. The box plot visually represents the maximum and minimum values (shown at the end of the whiskers), the interquartile range (shown as the box), and the median (the line dividing the box into two parts). *p*-values are listed on the right. (**c**) Lollipop plot depicts the impact of each 3′-end formation complex component on the hazard ratio (HR) of HCC patients. The width of the line represents the 95% confidence intervals of each study, the dot on each line represents the HR of that study, and the size of the dot represents the “weight” of the study. *p*-values are listed on the right. HR is an index to judge whether the expression of a certain gene is a favorable or unfavorable factor for a patient’s survival. When the HR < 1 and *p* < 0.05, it indicates that high expression of the gene can lower the risk of death and improve patient survival. Conversely, when the HR > 1 and *p* < 0.05, it suggests that high expression of the gene increases the risk of death and reduces patient survival. (**d**) The correlation of each 3′-end formation complex component with the total circRNA level is analyzed by Spearman’s correlation test; the *p*-values are listed on the right.

**Figure 2 cancers-15-04057-f002:**
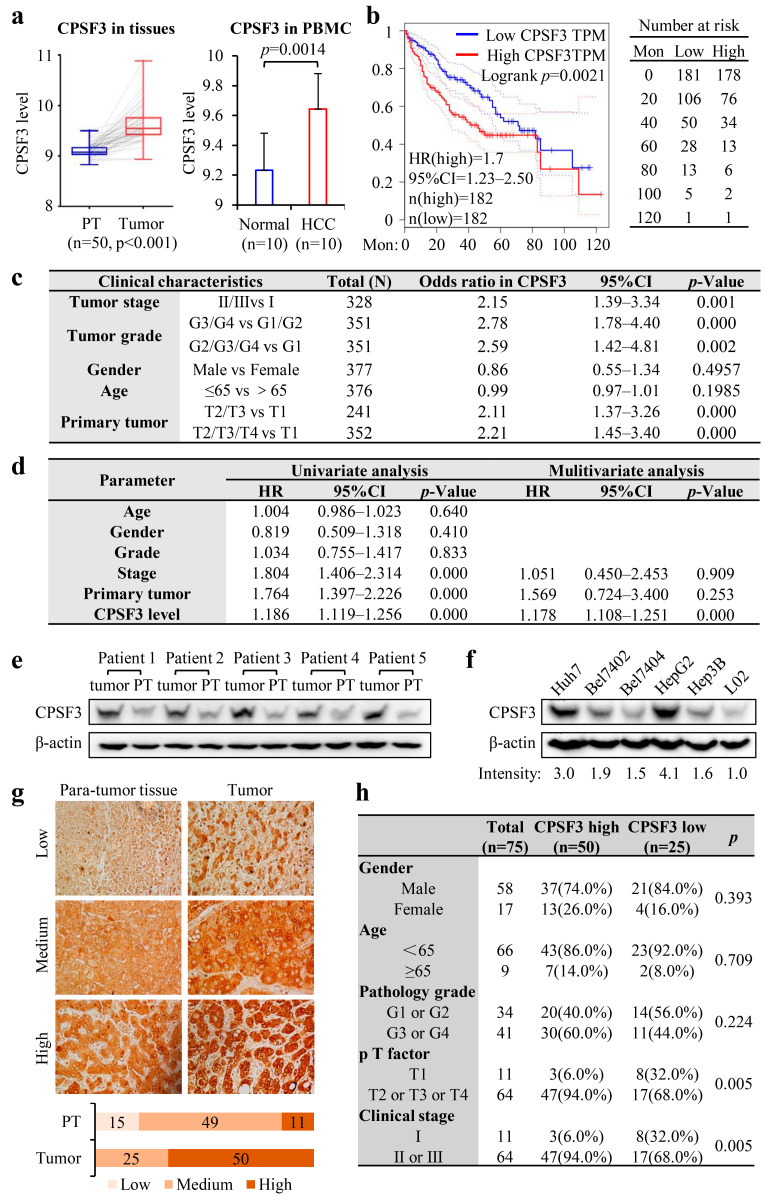
CPSF3 is overexpressed in HCC, and a high CPSF3 level indicates a poor prognosis in HCC. (**a**) The left panel shows that CPSF3 mRNA levels are significantly higher in HCC tissues compared to PTs. The right panel shows that CPSF3 mRNA levels are significantly higher in PBMC samples obtained from HCC patients compared to those obtained from healthy individuals. Error bars indicate standard deviations. (**b**) Kaplan–Meier survival curve analysis reveals that patients with high CPSF3 levels have a shorter OS. The solid line represents the survival curve, while the dashed lines above and below represent the 95% confidence interval. (**c**) Logistic regression analysis demonstrates significant associations between CPSF3 and clinicopathological features in HCC patients. (**d**) Univariate and multivariate analyses are performed to evaluate the prognostic significance of CPSF3 levels in comparison to other clinical parameters using the Cox regression model. (**e**) Western blot shows the protein levels of CPSF3 in HCC tissues and their corresponding PTs from five patients. (**f**) Western blot analysis demonstrates higher CPSF3 protein expression in HCC cell lines compared to normal hepatocytes. (**g**) Immunohistochemical staining of the tissue microarray revealing the expression of CPSF3 in HCC tissues and their PTs. The slide is photographed through an optical microscope, and the images are magnified by a 10× eyepiece and a 10× objective lens. The upper panel lists the representative images, and the bottom panel shows the statistical analysis. (**h**) The association between CPSF3 expression levels and the clinicopathological features is analyzed using the χ^2^ test. The uncropped blots are shown in Appendix A.

**Figure 3 cancers-15-04057-f003:**
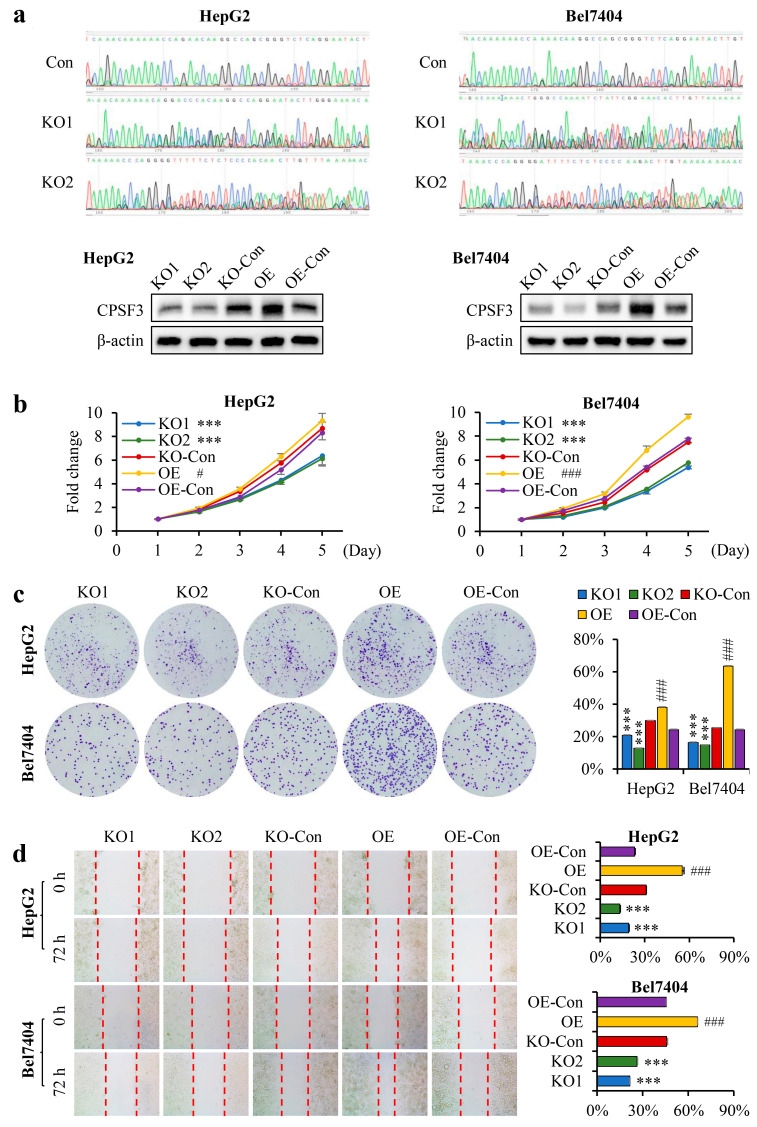
CPSF3 promoted the proliferation and migration of HCC cells. (**a**) Genome sequencing analysis of CPSF3 in HepG2 and Bel7404 KO cell lines reveals nested peaks, suggesting sequence disorder. Different colors represent different bases: adenine (green), guanine (black), cytosine (blue), thymine (red). Western blot results indicate that CPSF3 protein expression is decreased in the KO cell lines of HepG2 and Bel7404 and increases in the OE cell lines. (**b**) The CCK8 assay results (*n* = 5) reveal increased cell growth in CPSF3 OE cells and decreased growth in CPSF3 KO cells compared to their respective controls. (**c**) The colony formation assay demonstrates that CPSF3-OE cells form more colonies than pcDNA3 control cells, while CPSF3-KO cells form fewer colonies than lentiCRISPR v2 control cells. The culture dishes are imaged using a regular camera without magnification. The left panel shows the representative images, and the right panel shows the statistical analysis (*n* = 3). (**d**) The cell scratch assay results show that CPSF3-OE cells had faster healing rates, while CPSF3-KO cells had slower healing rates. The culture dishes were imaged using an optical microscope equipped with a 10× eyepiece and a 10× objective lens. The red dashed lines delineate the forefront of cellular migration. The left panel shows the representative images, and the right panel shows the statistical analysis (*n* = 3). In this figure, KO1 and KO2 indicate the knockout cells, which are compared to lentiCRISPR v2 vector-transfected cells; *** *p* < 0.001. OE denotes the overexpressing cells, which are compared to the pcDNA3 vector-transfected cells; # *p* < 0.05, ### *p* < 0.001. Error bars indicate standard deviations. The uncropped blots are shown in Appendix A.

**Figure 4 cancers-15-04057-f004:**
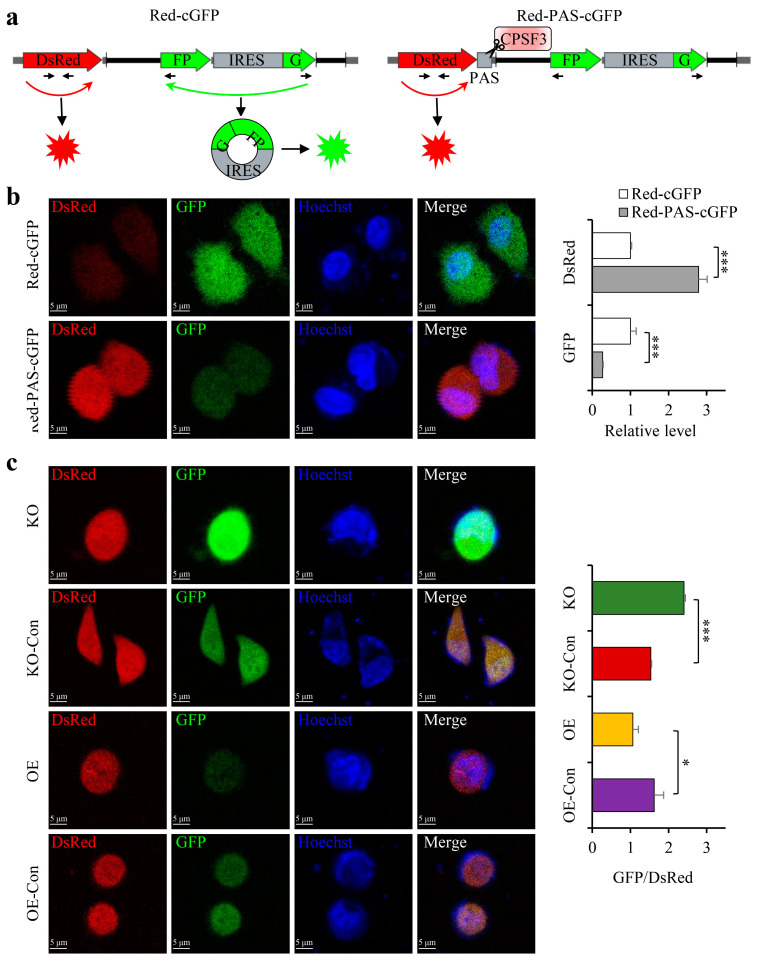
The CPSF3 and PAS elements are critical for circRNA cyclization. (**a**) Schematic diagram of Red-cGFP and Red-PAS-cGFP reporter plasmids. The arrows indicate the positions of primers for PCR. (**b**) A preference for linear RNA was observed in plasmids with PAS, while plasmids without PAS preferentially produced circRNA. The left panel shows the fluorescence of DsRed and circ-GFP detected by confocal microscopy. The right panel shows RNA levels of DsRed and circ-GFP measured by real-time PCR. (**c**) CPSF3 promoted the shift of pre-mRNA from circRNA to linear mRNA. Red-PAS-cGFP plasmids were transfected into cells with varying CPSF3 levels. The left panel shows the fluorescence of DsRed and circ-GFP while the right panel shows the ratio of circ-GFP to Ds-Red mRNA. In the microscopy image, the scale bar = 5 μm, and nuclei were counterstained with DAPI. Error bars indicate standard deviations (*n* = 3). * *p* < 0.05, *** *p* < 0.001.

**Figure 5 cancers-15-04057-f005:**
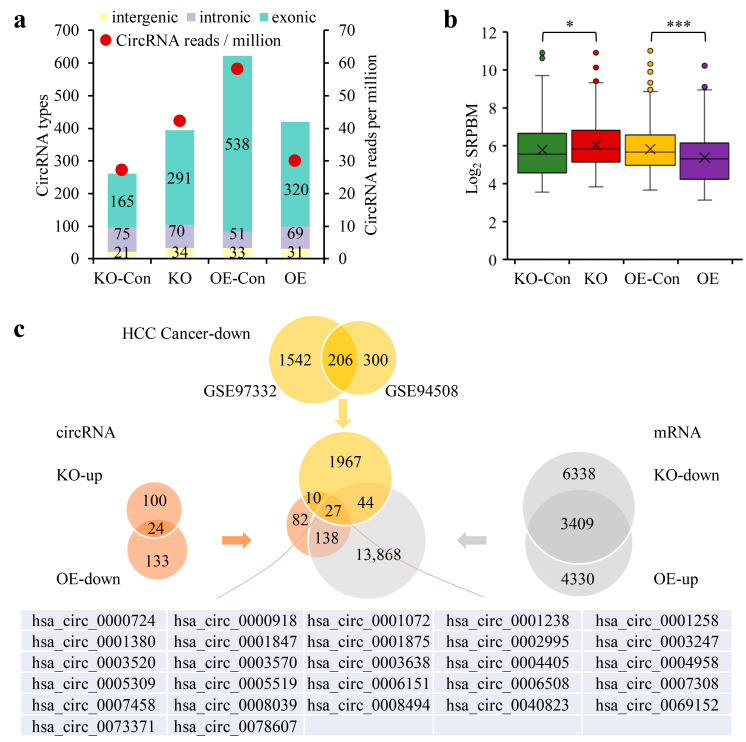
RNA sequencing data showed that CPSF3 reduced circRNA biogenesis in HCC cells. (**a**) Number and types of circRNAs in HCC cells with varied CPSF3 expression. (**b**) Box plot depicting average circRNA levels in HCC cells with different CPSF3 levels. (**c**) Venn diagram representing the gene screening process. CircRNAs negatively associated with CPSF3, mRNAs positively associated with CPSF3, and circRNAs that were reduced in HCC were considered candidate genes. * *p* < 0.05, *** *p* < 0.001.

**Figure 6 cancers-15-04057-f006:**
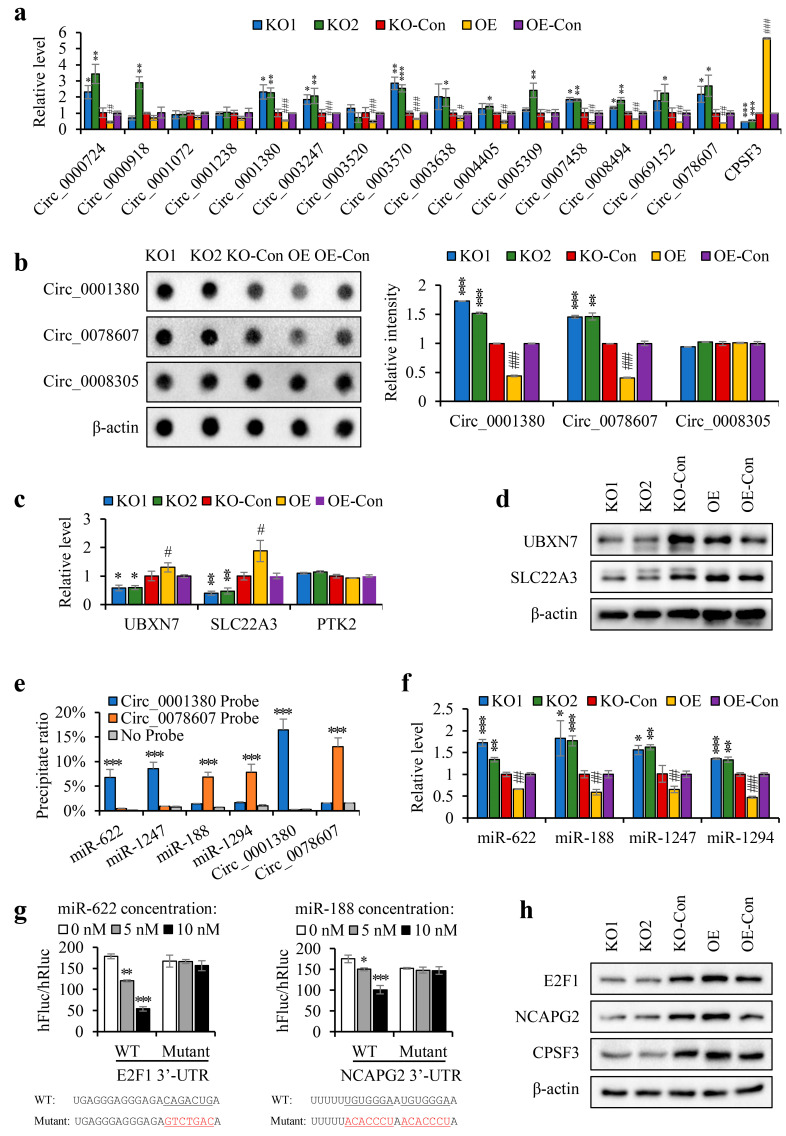
CPSF3 promoted the shift of pre-mRNA from circRNA to linear mRNA and disrupted miRNA-mediated gene silencing. (**a**) The expression levels of the screened circRNAs were detected by real-time PCR. (**b**) A dot blot assay was performed to determine the expression levels of circRNAs in CPSF3-KO and CPSF3-OE cells. (**c**,**d**) Real-time PCR and western blotting were conducted to detect the expression of linear mRNAs and proteins corresponding to circRNAs in CPSF3-KO and CPSF3-OE cells. (**e**) The biotin-labeled RNA pull-down assay results confirm the binding of circRNA to miRNA. (**f**) Real-time PCR analysis reveals the levels of miRNA in CPSF3-KO and CPSF3-OE cells. (**g**) The luciferase assay was conducted to confirm the silencing effect of miRNAs on target genes. (**h**) The protein levels of miRNA target genes and CPSF3 in cell lines were measured using western blot analysis. β-actin was used as the internal control. For (**a**–**c**,**f**), CPSF3-KO cells were compared to HepG2 cells (* *p* < 0.05, ** *p* < 0.01, *** *p* <0.001), and CPSF3-OE cells were compared to pcDNA3 vector-transfected Bel7404 cells (# *p* < 0.05, ## *p* < 0.01, ### *p* < 0.001). For (**e**), the co-precipitation results were compared to the group without probe. For (**g**), miRNA-treated groups were compared to no-treated groups (* *p* < 0.05, ** *p* < 0.01, *** *p* <0.001). Error bars indicate standard deviations (*n* = 3). The uncropped blots are shown in Appendix A.

**Figure 7 cancers-15-04057-f007:**
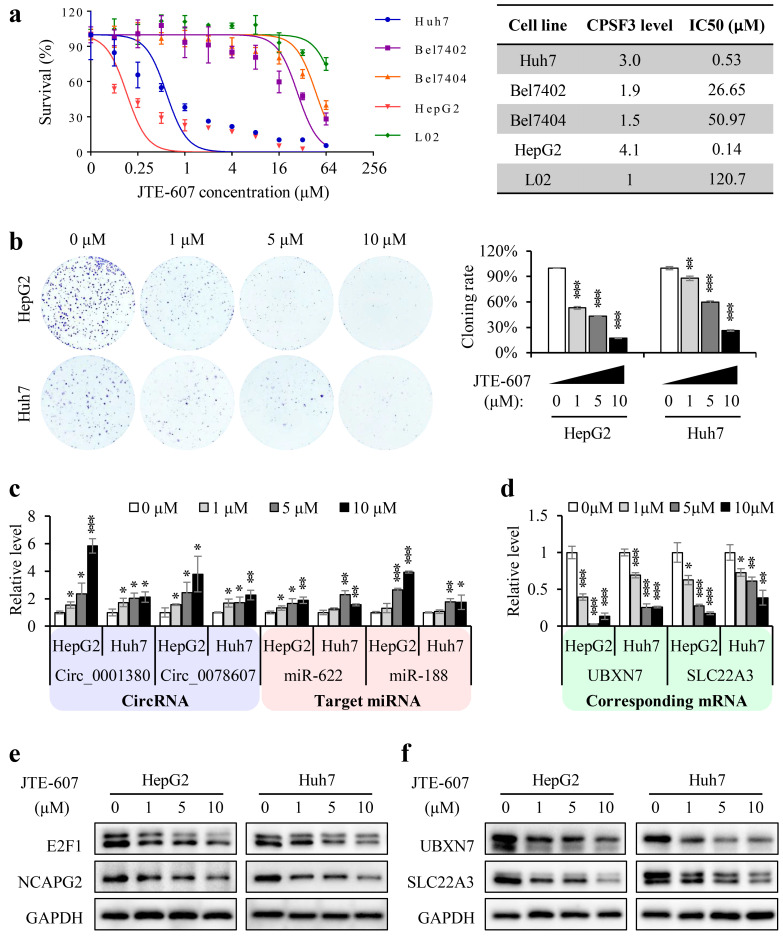
JTE-607 inhibited the proliferation of HCC cells. (**a**) The IC50 of JTE-607 were determined in HCC cell lines and a normal hepatocyte line. (**b**) JTE-607 inhibited the colony formation ability of HepG2 and HuH7 cells in a concentration-dependent manner. The color depth of the bar represents the concentration of JTE-607; the darker the color, the higher the concentration of JTE-607. (**c**,**d**) Real-time PCR showed that JTE-607 increased the expression levels of circRNAs and their target miRNAs in a concentration-dependent manner and decreased the levels of corresponding mRNAs. (**e**,**f**) Western blot revealed that JTE-607 caused a concentration-dependent decrease in the expression of miRNA target proteins and circRNA counterpart gene proteins. Error bars indicate standard deviations (*n* = 3). * *p* < 0.05, ** *p* < 0.01, *** *p* < 0.001. The uncropped blots are shown in Appendix A.

**Figure 8 cancers-15-04057-f008:**
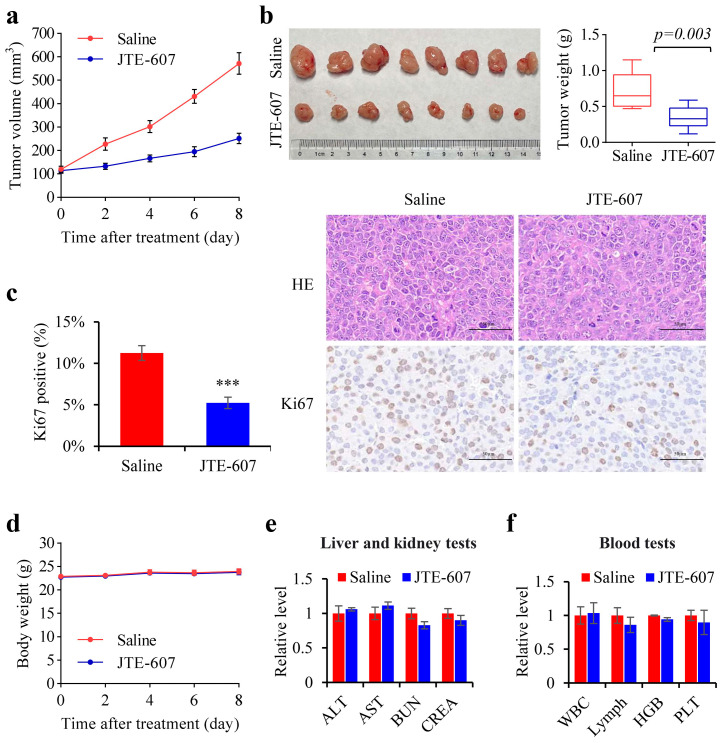
Efficacy and safety of JTE-607 in tumor-bearing nude mice. (**a**) Tumor growth curves of nude mice during JTE-607 administration. (**b**) Tumors were photographed and weighed after excision. (**c**) Above is the hematoxylin and eosin staining and below is the immunohistochemical staining results for Ki67. Scale bar, 50 µm. (**d**) Changes in the body weight of nude mice during JTE-607 treatment. (**e**) Serum levels of ALT, AST, BUN, and CREA in tumor-bearing nude mice were measured two days after the last treatment. (**f**) Blood cell parameters (WBC, lymph, HCB, and PLT) in tumor-bearing nude mice two days after the last treatment. Error bars represent standard deviations (*n* = 8). *** *p* < 0.001.

## Data Availability

The data presented in this study are available in the article or Appendix A.

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
