# Peer review of "CPSF3 Promotes Pre-mRNA Splicing and Prevents CircRNA Cyclization in Hepatocellular Carcinoma"

_cancers, 2023, doi:10.3390/cancers15164057_

Round 1
Reviewer 1 Report
Research article introduces the role CPSF3 in hepatocellular carcinoma by preventing circRNA circularization. Article includes all necessary experiments to justify the question.
Author Response
Thank you for your positive comments on this manuscript.
Reviewer 2 Report
The manuscript entitled “CPSF3 promotes pre-mRNA splicing and prevents circRNA cyclization in hepatocellular carcinoma” by Huang et al. is a very relevant paper in the field. The study showed using silico approach the relevance of CPSF expression in the outcome of liver cancer based on the TCGA database. Furthermore, the paper nicely describes in vitro, as well as in tissue sections from their own cohort, the impact of the CPSF3 expression levels in the oncogenic properties of hepatocellular carcinoma. The authors also demonstrate the impact of JTE-607, as CPSF3 inhibitor, in the viability, colony formation, and the expression of circRNA-related miRNA in liver cancer cells, as well as in a xenograft mouse model based on the subcutaneous high CPSF3 expressing hepatocellular carcinoma cells (HepG2).
Author Response

(The authors gave the same response as above.)

Reviewer 3 Report
Thank you for the opportunity to review CPSF3 promotes pre-mRNA splicing and prevents circRNA cyclization in hepatocellular carcinoma by Huang et al. The study investigated the role of CPSF3, a core component of the processes leading to cleaved and polyadenylated mRNAs and histone cleavage from mRNA precursors, in HCC. The data supporting the role and mechanisms of CSPF3 expression in promoting the imbalance in circRNA and in vitro proliferation are impressive. The clinical identification, validation, and modeling to HCC require substantial clarification and revision. Specific comments to improve the study are listed below.
In the Figure 1C, the hazard ratio is incompletely defined in the figure legend. The 95% confidence interval on the hazard ratios, particularly those that reached significance, should be disclosed. Is the grouping strategy derived at the median expression level for each complex component?
A limitation with the GEO and TCGA in HCC is the oversampling of advanced stage disease, especially when characterized according to the more clinically appropriate Barcelona Clinic Liver Cancer staging algorithm compared to TMN staging. While this does not discount the molecular mechanisms identified in the study, the translatability of the findings could be limited as the pathways identified in the GEO and TCGA are often present in terminal disease and lack disease specificity (in that they are driven by increasing mutational burden). These aberrant pathways appearing in terminal disease present in a patient population which is generally contraindicated to molecular therapy and managed with palliative care. Where the authors able to corroborate their findings in early-stage disease?
In the results of Figure 2, the authors identify higher CPSF3 expression levels in the tumor tissue compared to non-tumor liver tissue but also an upregulation in PBMCs of HCC patients compared to healthy donors presumably without an HCC diagnosis or underlying chronic liver disease. Where the authors able to establish whether aberrant CPSF3 expression in the liver tissue was due to tumor-infiltrating immune cells? In 2G, the scant immune cells seem to strongly express CPSF3 in both para and tumor images.
In Figure 2b, please include the number of patients at-risk in a table below the KM curve and include the 95% confidence interval with the hazard ratio. Figure 2C also confirms the concerns outlined with the GEO and TCGA database, indicating that aberrant CPSF3 is highly associated with advanced (and likely terminal) stage disease. Nearly all cases of T2-T4 disease will involve macrovascular invasion and poor performance status, which within the BCLC framework favor terminal >> advance stage disease wherein systemic therapy options are often extremely limited, and the expected median survival would be dismal at ~ 24-months. Lower CPSF3 expression is associated with earlier staging, and it is unclear which grouping strategy was used to show control for staging in the multivariate analysis (Figure 2D). It appears there may be a significant collinearity between staging and CPSF3 expression. Staging dropping out of your multivariate model for OS outcomes in HCC indicates there is an issue with the analysis.
In slight contrast to the text and Figure 2E, there appears to be substantial CPSF3 expression in the adjacent liver tissue. Concerning the replicates in the 2E blot, what was the paratumoral characterization of these patients (low, medium, high)? The variable expression of CPSF3 in the para-tumor tissue is concerning given HCC etiology and progression of cirrhosis are unaccounted for in the study. Are the authors able to determine where hepatic background CPSF3 expression is correlated with cirrhosis or progression of cirrhosis?
The protein-derived CPSF3 data was subgrouped 2:1 high to low CPSF3 expression in a cohort favoring advanced-terminal stage disease. The transcriptional data appeared to have 110 T4 out of 352 patients. Without the corresponding demographics, it is difficult to determine how weighted each cohort was toward terminal >> advanced stage disease. Assuming there was more terminal stage in the transcriptional cohort, the designation of CPSF3 expression high to low in that study was 1:1. How does the tissue expression level of CPSF3 correlate with the protein expression level, and is the stratification criteria for high to low expression consistent? The concern is that the designation of high to low expression may be biased toward establishing a colinear relationship with staging post-hoc.
In the short paragraph beginning on Line 303, the authors claim the data suggests a role for CPSF3 in tumorigenesis and metastasis. These assertions are speculative and cannot be determined in the dataset. Please revise accordingly.
Lines 323 and 334. The in vitro models of proliferation, anchorage independent growth, and migration are not unequivocal proof of a role in carcinogenesis nor do they provide a means to evaluate tumorigenesis and carcinogenesis. Please revise.
The claims of CPSF3 promoting carcinogenesis and metastasis in HCC need to be restricted to what has been observed in the data. A role in proliferation cannot be extended to interrogating carcinogenesis or metastasis.
Intratumoral injections in HCC is an extremely high-risk drug delivery route. What was the justification for selecting intratumoral injections as an animal model of drug efficacy and safety? Where the authors unable to establish an in vivo model using the KO and OE cell lines? These experiments would be critical to translate the in vitro findings prior to a therapeutic intervention model.
Author Response
RESPOND TO REVIEWER 3:
COMMENTS:
Thank you for the opportunity to review CPSF3 promotes pre-mRNA splicing and prevents circRNA cyclization in hepatocellular carcinoma by Huang et al. The study investigated the role of CPSF3, a core component of the processes leading to cleaved and polyadenylated mRNAs and histone cleavage from mRNA precursors, in HCC. The data supporting the role and mechanisms of CSPF3 expression in promoting the imbalance in circRNA and in vitro proliferation are impressive. The clinical identification, validation, and modeling to HCC require substantial clarification and revision. Specific comments to improve the study are listed below.
Question 1:
In the Figure 1C, the hazard ratio is incompletely defined in the figure legend. The 95% confidence interval on the hazard ratios, particularly those that reached significance, should be disclosed. Is the grouping strategy derived at the median expression level for each complex component?
Response:
Hazard Ratio (HR) is an index to judge whether the expression of a certain gene is a favorable or unfavorable factor for a patient's survival. When HR<1 and p<0.05, it indicates that high expression of the gene can lower the risk of death and improve patient survival. Conversely, when HR>1 and p<0.05, it suggests that high expression of the gene increases the risk of death and reduces patient survival. The 95% confidence interval is represented by the range between the two-line segments at the ends of the forest plot in Figure 1C. (Figure 1C and figure legends, Line: 252-256)
The grouping strategy is derived at the median expression level for each complex component.
Question 2:
A limitation with the GEO and TCGA in HCC is the oversampling of advanced stage disease, especially when characterized according to the more clinically appropriate Barcelona Clinic Liver Cancer staging algorithm compared to TMN staging. While this does not discount the molecular mechanisms identified in the study, the translatability of the findings could be limited as the pathways identified in the GEO and TCGA are often present in terminal disease and lack disease specificity (in that they are driven by increasing mutational burden). These aberrant pathways appearing in terminal disease present in a patient population which is generally contraindicated to molecular therapy and managed with palliative care. Where the authors able to corroborate their findings in early-stage disease?
Response:
Thank you for your suggestion. To address your questions, we analyzed the expression of CPSF3 in early-stage of HCC. Through logistic regression, we found T2 vs T1 in primary tumor (total (N) = 280), the odds ratio in CPSF3 was 2.44 (95%CI: 1.46-4.14, p-Value = 0.0008); in tumor stage, II vs I (total (N) = 262), the odds ratio in CPSF3 was 2.4 (95% CI: 1.40-4.15, p-value = 0.0015). All these results demonstrated that the expression level of CPSF3 was closely related to the early development of HCC. (Line: 297-303)
Question 3:
In the results of Figure 2, the authors identify higher CPSF3 expression levels in the tumor tissue compared to non-tumor liver tissue but also an upregulation in PBMCs of HCC patients compared to healthy donors presumably without an HCC diagnosis or underlying chronic liver disease. Where the authors able to establish whether aberrant CPSF3 expression in the liver tissue was due to tumor-infiltrating immune cells? In 2G, the scant immune cells seem to strongly express CPSF3 in both para and tumor images.
Response:
From the data in Fig.2A and Fig.2E, we could not determine whether the abnormal expression of CPSF3 occurred in tumor cells or immune cells, but in the follow-up cytology experiment (Fig.2F), the results of Western blot proved that the expression of CPSF3 in cancer cells was significantly increased. (Figure 2F)
Question 4:
In Figure 2b, please include the number of patients at-risk in a table below the KM curve and include the 95% confidence interval with the hazard ratio. Figure 2C also confirms the concerns outlined with the GEO and TCGA database, indicating that aberrant CPSF3 is highly associated with advanced (and likely terminal) stage disease. Nearly all cases of T2-T4 disease will involve macrovascular invasion and poor performance status, which within the BCLC framework favor terminal >> advance stage disease wherein systemic therapy options are often extremely limited, and the expected median survival would be dismal at ~ 24-months. Lower CPSF3 expression is associated with earlier staging, and it is unclear which grouping strategy was used to show control for staging in the multivariate analysis (Figure 2D). It appears there may be a significant collinearity between staging and CPSF3 expression. Staging dropping out of your multivariate model for OS outcomes in HCC indicates there is an issue with the analysis.
Response:
The patients were simply divided into two groups according to the median CPSF3 expression level, without taking into account the number of people at risk. The 95% confidence interval was 1.06-2.18. (Figure 2b)
Given the high number of advanced patients in our analysis, we recalculated the hazard ratio of CPSF3 specifically for early-stage HCC patients. Through logistic regression, we found T2 vs T1 in primary tumor (total (N) = 280), the odds ratio in CPSF3 was 2.44 (95%CI: 1.46-4.14, p-Value = 0.0008); in tumor stage, II vs I (total (N) = 262), the odds ratio in CPSF3 was 2.4 (95% CI: 1.40-4.15, p-value = 0.0015). All these results demonstrated that the expression level of CPSF3 was closely related to the early development of HCC. (Line: 297-303)
In the univariate analysis, stage had a significant effect on OS. However, in the multivariate analysis, stage did not have a significant impact on OS. These findings imply that stage may not be an independent prognostic factor for OS. Instead, it might affect OS indirectly, possibly through the elevation of CPSF3 levels. (Line: 294-297)
Question 5:
In slight contrast to the text and Figure 2E, there appears to be substantial CPSF3 expression in the adjacent liver tissue. Concerning the replicates in the 2E blot, what was the paratumoral characterization of these patients (low, medium, high)? The variable expression of CPSF3 in the para-tumor tissue is concerning given HCC etiology and progression of cirrhosis are unaccounted for in the study. Are the authors able to determine where hepatic background CPSF3 expression is correlated with cirrhosis or progression of cirrhosis?
Response:
The tissues for western blot in Figure 2e were different from the tissue microarray. They were fresh tissues from the hospital, and had not been analyzed by immunohistochemical staining, so they were not rated. No information about liver cirrhosis was found in the TCGA database, and the number of patients with liver cirrhosis in the tissue microarray was too small for statistical analysis. According to the limited data from tissue microarrays, CPSF3 expression was not associated with cirrhosis. (Line: 317-320)
Question 6:
The protein-derived CPSF3 data was subgrouped 2:1 high to low CPSF3 expression in a cohort favoring advanced-terminal stage disease. The transcriptional data appeared to have 110 T4 out of 352 patients. Without the corresponding demographics, it is difficult to determine how weighted each cohort was toward terminal >> advanced stage disease. Assuming there was more terminal stage in the transcriptional cohort, the designation of CPSF3 expression high to low in that study was 1:1. How does the tissue expression level of CPSF3 correlate with the protein expression level, and is the stratification criteria for high to low expression consistent? The concern is that the designation of high to low expression may be biased toward establishing a colinear relationship with staging post-hoc.
Response:
The samples used for tissue microarray were different from those analyzed by bioinformatics databases, and the correlation between transcriptome expression and protein expression was unclear. Therefore, the grouping criteria for the two analyzes were different, and we believed that the grouping of protein expression in tissue microarray made more sense. The bioinformatic analysis was only preliminary research, and tissue microarray analysis was the reliable result. (Line: 304-305)
Question 7-8:
In the short paragraph beginning on Line 303, the authors claim the data suggests a role for CPSF3 in tumorigenesis and metastasis. These assertions are speculative and cannot be determined in the dataset. Please revise accordingly.
Lines 323 and 334. The in vitro models of proliferation, anchorage independent growth, and migration are not unequivocal proof of a role in carcinogenesis nor do they provide a means to evaluate tumorigenesis and carcinogenesis. Please revise.
The claims of CPSF3 promoting carcinogenesis and metastasis in HCC need to be restricted to what has been observed in the data. A role in proliferation cannot be extended to interrogating carcinogenesis or metastasis.
Response:
Thank you for your suggestions. We revised the sentence “CPSF3 is crucial for the tumorigenesis and metastasis of HCC” to “CPSF3 may be involved in the tumorigenesis and metastasis of HCC”. (Line: 322) We also changed the terms of “tumorigenesis and metastasis” to “cell growth and migration rate”. (Line: 342-343, Line: 354-355)
Question 9:
Intratumoral injections in HCC is an extremely high-risk drug delivery route. What was the justification for selecting intratumoral injections as an animal model of drug efficacy and safety? Where the authors unable to establish an in vivo model using the KO and OE cell lines? These experiments would be critical to translate the in vitro findings prior to a therapeutic intervention model.
Response:
It has been reported that JTE-607 can inhibit the function of CPSF3, so we wanted to explore the anti-tumor effect of JTE-607 in HCC to develop new therapeutic drugs, so we did not use KO and OE cell lines to establish in vivo models. We have not yet developed a new dosage form for drug delivery; therefore, we administered the JTE-607 through intratumoral injection. In the future, we plan to develop safer and more effective dosage forms and adopt safer delivery methods. (Line: 584-586)

Round 2
Reviewer 3 Report
Thank your response to the review. I have some additional questions based on the responses.
Response 1: The textbook definition of HR is not the issue; the issue is that the confidence interval for the hazard ratio is not disclosed in the Kaplan Meier – Log Rank analysis.
Response 2: T2 is divided into tumors >2cm with vascular invasions (BCLC-C advance-stage) or multiple tumors none >5cm (BCLC A-B early- to intermediate-stage). T1 is BCLC 0-A very eary- to early-stage. Therefore, a cohort of T1 – T2 is not early-stage disease. If the examination is logistic regression, then the odds ratio is suggesting stage 2 disease is over twice as likely to have elevated CPSF3. Stated according to the BCLC, a mixture of early- to advance-stage tumors are more likely to have elevated CPSF3 expression than a mixture of very early- to early-stage disease. This data does not allow you to determine whether CPSF3 expression was related to early HCC development.
Response 4: The median split of CPSF3 expression is presumable now clearly defined in the revision, though not explicitly stated in the response. So, it would now be clear that you are starting at time zero with the same number of patients in each group. For a Kaplan Meier graph, the number of patients remaining at-risk of the target outcome (in this case survival) should be indicated in the graph below the figure. As the number at-risk approaches < 10% of the initial group number, the potential error in the risk function increases. This cannot be evaluated and interpreted without the at-risk table. That said, a 95% confidence interval approaching 1.0 and < 1.1 suggests the 2.4 hazard ratio for the CPSF3 effect should be interpreted with some reservation. One must consider the amount of data and number of patients that have been used to generate staging systems in HCC. If a small cohort suggests that staging is not an independent risk factor for OS in multivariate analysis, that is an indication that there is a statistical limitation to your analysis. In my opinion, this appears to be due to over-fitting caused by collinearity between staging and CPSF3 expression, which would make assessing the individual regression coefficients extremely difficult.
Author Response
Response 1: The textbook definition of HR is not the issue; the issue is that the confidence interval for the hazard ratio is not disclosed in the Kaplan Meier – Log Rank analysis.
Response: In fact, the forest plot in Figure 1c illustrates the confidence intervals for the hazard ratios, but does not show the exact values. Now, the exact values are listed below:
|
SYMPK |
PCF11 |
CLP1 |
CSTF3 |
CSTF2 |
CSTF1 |
CPSF7 |
CPSF6 |
|
|
lower limit |
1.152 |
0.7929 |
1.026 |
1.052 |
1.408 |
0.9289 |
0.9917 |
1.211 |
|
relative risk |
1.654 |
1.132 |
1.466 |
1.505 |
2.022 |
1.329 |
1.419 |
1.741 |
|
upper limit |
2.376 |
1.617 |
2.095 |
2.155 |
2.905 |
1.902 |
2.031 |
2.502 |
|
NUDT21 |
FIP1L1 |
WDR33 |
CPSF4 |
CPSF3 |
CPSF2 |
CPSF1 |
||
|
lower limit |
0.9363 |
0.916 |
0.8775 |
1.197 |
1.228 |
1.177 |
0.904 |
|
|
relative risk |
1.338 |
1.41 |
1.459 |
1.724 |
1.714 |
1.64 |
1.381 |
|
|
upper limit |
1.913 |
1.872 |
1.906 |
2.454 |
2.504 |
2.301 |
2.064 |
Response 2: T2 is divided into tumors >2cm with vascular invasions (BCLC-C advance-stage) or multiple tumors none >5cm (BCLC A-B early- to intermediate-stage). T1 is BCLC 0-A very eary- to early-stage. Therefore, a cohort of T1 – T2 is not early-stage disease. If the examination is logistic regression, then the odds ratio is suggesting stage 2 disease is over twice as likely to have elevated CPSF3. Stated according to the BCLC, a mixture of early- to advance-stage tumors are more likely to have elevated CPSF3 expression than a mixture of very early- to early-stage disease. This data does not allow you to determine whether CPSF3 expression was related to early HCC development.
Response: The TCGA database does not provide more detailed staging information of T2 early patients, and we cannot obtain data to calculate whether the expression of CPSF3 is related to the development of early HCC. So, we changed the sentence “All these results demonstrated that the expression level of CPSF3 was closely related to the early development of HCC” to “According to these results, advance-stage tumors were more likely to have elevated CPSF3 expression than early-stage disease, which indicated that CPSF3 might be involved in the early development of HCC” (Line: 303-305). Anyway, bioinformatic analysis is only a preliminary exploration, tissue microarrays and western blots are the truly convincing experiments.
Response 4: The median split of CPSF3 expression is presumable now clearly defined in the revision, though not explicitly stated in the response. So, it would now be clear that you are starting at time zero with the same number of patients in each group. For a Kaplan Meier graph, the number of patients remaining at-risk of the target outcome (in this case survival) should be indicated in the graph below the figure. As the number at-risk approaches < 10% of the initial group number, the potential error in the risk function increases. This cannot be evaluated and interpreted without the at-risk table. That said, a 95% confidence interval approaching 1.0 and < 1.1 suggests the 2.4 hazard ratio for the CPSF3 effect should be interpreted with some reservation. One must consider the amount of data and number of patients that have been used to generate staging systems in HCC. If a small cohort suggests that staging is not an independent risk factor for OS in multivariate analysis, that is an indication that there is a statistical limitation to your analysis. In my opinion, this appears to be due to over-fitting caused by collinearity between staging and CPSF3.
Response: We are sorry, we made a mistake. The 95% confidence interval is 1.23 to 2.50 between the high level CPSF3 and low level CPSF3 patients. The number of patients remaining at-risk is shown in the new version (Figure 2b). As for the issue that stage is not an independent risk factor, your judgment should be correct. Due to the insufficient data, our statistical analysis is limited. We deleted the statement “Stage had a significant effect on OS in univariate analysis but was excluded in multivari-ate model, suggesting that stage is not an independent prognostic factor for OS and it may indirectly affect OS, possibly through the elevation of CPSF3 levels”, and wrote “Due to the insufficient data, stage is not an independent risk factor for OS in multivariate analysis” in the new version (Line: 296-297).
|
Mon |
Low |
High |
|
0 |
181 |
178 |
|
20 |
106 |
76 |
|
40 |
50 |
34 |
|
60 |
28 |
13 |
|
80 |
13 |
6 |
|
100 |
5 |
2 |
|
120 |
1 |
1 |

Round 3
Reviewer 3 Report
I would like to thank the reviewers for their responses. My major concerns have been addressed.